# Surgical Strategies for Combined Hepatocellular-Cholangiocarcinoma (cHCC-CC)

**DOI:** 10.3390/cancers15030774

**Published:** 2023-01-26

**Authors:** Marcus Bahra, Ali Yahyazadeh

**Affiliations:** Department of Surgical Oncology and Robotics, Krankenhaus Waldfriede, Lehrkrankenhaus der Charité Argentinische Alle 40, 14163 Berlin, Germany

**Keywords:** combined hepatocellular-cholangiocarcinoma (cHCC-CC), surgical, liver

## Abstract

**Simple Summary:**

Combined hepatocellular-cholangiocarcinoma is a rare liver tumor with hepatocellular as well as cholangiocellular differentiation patterns. In this review we give an insight of the pathological as well as clinical presentations and surgical treatment options to optimize the knowledge of this tumor entity.

**Abstract:**

Combined hepatocellular–cholangiocarcinoma (cHCC-CC) is a tumor entity presenting features of hepatocellular and cholangiocellular epithelial differentiation. Due to the likeness between cHCC-CC, HCC and CC, accurate pretherapeutical diagnosis is challenging and advanced stages are prevalent. Radical oncological surgery is the only curative therapeutical option in patients with cHCC-CC. To reach this goal a profound understanding of this rare liver tumor is crucial. Factors such as clinicopathological characteristics, growth patterns and biological behavior are of central importance. To explore onco-surgical strategies and aspects for complete resection of cHCC-CC and to answer important key questions, an extensive review of the literature was conducted to answer the following questions: What are the best surgical options? Is there a significance for nonanatomical resections? Is there a prognostic value of concomitant lymphadenectomy? What about multimodal concepts in local advanced cHCC-CC? The role of minimally invasive liver surgery (MILS) including the role of robotic liver surgery for cHCC-CC will be discussed. While liver transplantation (LT) is standard for patients with unresectable HCC, the role of LT in cHCC-CC patients is still controversial. How can patients with high risk for early tumor recurrence be identified to avoid aggressive surgical treatment without clinical benefit? The comprehensive understanding of this challenging liver tumor will help to improve future treatment options for these patients.

## 1. Introduction

Combined hepatocellular–cholangiocarcinoma (cHCC-CC) or mixed hepatocellular–cholangiocarcinoma were regarded as “collision tumors“ in the past. To date, the term combined HCC-CC is common. However, a stringent definition of this entity is still under discussion [1]. The relationships between combined HCC-CC and either HCC or CC are not even clear yet. Janargin et al. postulated that demographic and clinical features are most similar to those of patients with cholangiocarcinoma [2]. On the other hand, it has been proposed to view cHCC-CC more as a kind of variant of HCC with cholangiocellular features [3]. Overall survival data of the past suggested that the clinical features of HCC-CC are more akin to HCC than to cholangiocarcinoma [4]. However, other studies found overall survival rates for combined HCC-CC to be shorter than those for HCC [5]. Based on different biological and pathological behaviour, the surgical concept is of central importance. How can this rare entity be treated? Should it be operated like cholangiocarcinomas or like hepatocellular carcinomas? Or is there the need for a completely new strategy for this tumor?

## 2. Pathological Characteristics

Since the first description by Allen and Lisa in 1949, several descriptions and subtypes of the tumor were established [6]. In 1985, Goodman et al. revised the classification with a new description of three types of cHCC-CC: the collision type, the transitional type and the fibrolamellar HCC with mucin-producing pseudoglands [7]. This classification, however, has certain similarities to intrahepatic cholangiocarcinomas which also shows distinct histological diversity. In the past, it was primarily assumed that cholangiocarcinomas derive from the epithelial cells of the biliary tree. Recent findings suggest that cholangiocarcinomas can alternatively originate from hepatic progenitor cells [8]. By comparing the development of cholangiocarcinomas and hepatocellular carcinomas, some interesting similarities can be observed during tumorigenesis. During the differentiation into human hepatocytes, four stages are the basis of the differentiation process. This includes embryonic stem cells, endoderm, liver progenitor cells and premature hepatocytes [9]. For that matter, both tumors, CC and HCC may originate from the same liver progenitor cell. However, the origin of combined hepatocellular–cholangiocarcinoma remains unclear. A recent study focused on the hepatic progenitor cells in the context of chronic inflammation. A multidrug resistance 2 knockout mouse model developing progressive liver disease including portal inflammation, sclerosing cholestatic hepatitis and ductal proliferation was used to induce a chronic inflammatory environment. The authors found that cHCC-CC could emerge from hepatic stem cells at later stages in a chronic inflammatory setting [10]. In other words, if mature hepatocytes have the ability to dedifferentiate into progenitor cells, they also are able to further undergo full malignant transformation into cHCC-CC. Taking this into account, it cannot be excluded that cHCC-CC may arise from dedifferentiated transformed hepatocytes.

For the clinical work-up, the 2019 WHO definition of cHCC-CC is important. The classification defines cHCC-CC as a primary liver carcinoma with obvious presence of both hepatocytic and cholangiocytic differentiation [11]. The conclusion for surgeons should be that cHCC-CC constitutes a distinct primary liver tumor with different molecular and clinical characteristics. It is neither HCC nor CC. It is an entity of its own.

## 3. Clinical Presentation and Preoperative Work-Up

Due to unspecific clinical symptoms such as fatigue, weight loss and abdominal discomfort, advanced tumoral stages are present mostly at the time of diagnosis. If a tumor mass is suspected of primary liver cancer showing elevated serum levels of both alpha-fetoprotein (AFP) and carbohydrate antigen 19.9 (CA 19-9), the diagnosis of cHCC-CC is very likely. In regard to imaging features, cHCC-CC shows a heterogeneous picture significantly overlapping with those of HCC and CC. The role of preoperative biopsy remains critical. Due to the heterogeneity of the tumor, different components of a radiologically heterogeneous tumor should be biopsied to get a sufficient diagnosis. However, the approach of performing multilocular biopsies is technically advanced and, in our point of view, risky and therefore difficult to realize in clinical routine. Gigante et al. suggest in their study that, due to lack of systematic biopsy for liver cancers, a significant number of cHCC-CC are currently misdiagnosed as HCC [12]. Radiological findings of the tumor are important to determine resectability and the scale of hepatectomy. Computertomography (CT) with the option of three-dimensional reconstruction is the central imaging modality of therapeutical decision-making. To get a comprehensive picture of the liver tumor, magnetic resonance imaging (MRI) provides important information about the presence of satellite nodules and liver fibrosis or cirrhosis. Most important, magnetic resonance cholangiopancreatography (MRCP) is needed to obtain detailed information about the extent of vascular invasion and an exact picture of the presence of biliary tumor involvement. Using the Child-Pugh score or the Model for End-Stage Liver Disease (MELD) score, significant information regarding the risk of postoperative hepatic insufficiency can be seen [13]. In addition, the future liver remnant (FLR) measured using CT-based liver volumetry is an important predictor of postoperative liver dysfunction in case of extended hepatectomies [14] Furthermore, precise assessment of the intrahepatic vascular systems is essential, to evaluate the risk of devascularization or venous congestion.

## 4. Surgical Strategies

### 4.1. Liver Resection

Surgical resection is the only curative therapeutical option for patients with malignant tumors of the liver. The knowledge about the surgical treatment of primary liver tumors is mainly based on experience from HCC and CC. A basic distinction can be made between anatomical and nonanatomical resections. Anatomical resections thereby are defined as resections of six or less segments based on the segmental anatomy of the liver. Nonanatomical resections comprise the removal of liver without paying attention on the segmental anatomy with tumor-free resection margins. However, for surgical resection of HCC, the indication has been expanded over the last two decades. Liver resection is indicated for all patients without metastatic disease and normal liver function. Patients with compensated cirrhosis and missing signs of portal hypertension should not be excluded from surgical therapy per se. When hepatectomy is considered in patients with reduced liver function, several aspects are important: the resection has to follow general oncological principles (R0-resection) and must also be realized in a way so as to ensure adequate liver function postoperatively. While in young patients with healthy livers an FLR of 20% is acceptable, for patients showing compensated liver cirrhosis with no signs of portal hypertension, a minimum FLR ratio of about 40% is needed for a safe resection [15]. In these patients, the use of perioperative portal vein embolization (PVE) is established as a considerable strategy [16]. Results of prospective clinical trials comparing PVE followed by hemihepatectomy to upfront surgery found that PVE significantly reduces postoperative complications in patients with underlying liver disease [17]. Furthermore, overall-survival and recurrence-free survival for patients with hepatectomy for HCC after PVE has been shown to be superior to those treated without PVE [18]. In extended liver tumors with the need for extended resection, two-staged hepatectomies as liver deprivation and ALPPS may also be useful techniques to overcome postoperative parenchymal shortage [19].

When treating even smaller tumors, the option of parenchyma-preserving anatomic resections should be discussed. This includes the right posterior sectioectomy, central hepatectomies, bisegmentectomies and monosegmentectomies, if a negative resection margin can be achieved. Compared to nonanatomical resections (wedge resections), several studies demonstrated that anatomical resections were associated with lower local recurrence rates and higher 5-year disease-free survival compared to nonanatomical resections [20]. An analysis of a nationwide Japanese database demonstrated no difference in overall survival and an improved disease-free survival after anatomical resections for HCC, indicating that minor liver resections for HCC are appropriate in a subgroup of patients with smaller tumors [20,21]. However, after stratification by tumor size, disease-free survival was significantly improved after anatomical resections for HCC with a diameter of 2 to 5 cm. Due to the low incidence of vascular invasion in HCC smaller than 2 cm, it seems to have no beneficial effect for anatomic resections in those patients [22]. What are the differences compared to the surgical therapy of intrahepatic cholangiocarcinoma? Macroscopically, mass forming types of intrahepatic CC have to be distinguished from periductal infiltrating and intraductal growth types. The most common type of CC is the mass-forming type that does not invade the main bile ducts. The periductal-infiltrating type grows along the length of the bile ducts. On the other hand, the intraductal growth type sprouts strictly intraluminal. The pathological picture can be a mix of all different types in the worst case [23]. As most patients present with advanced disease, more than 75% of patients require a hemihepatectomy or extended hepatectomy for tumor removal [24,25]. Resectability for CC follows the same principles as for HCC. However, due to the growth pattern with high risk of vascular invasion and satellite nodules, anatomical resections should be prioritized to ensure negative tumor margins. For that, nonanatomical resections or wedge resections may be less advantageous in CC. In contrast to HCC, staging laparoscopy should be discussed in patients with CC, multicentric disease and high CA 19-9 levels to preclude patients as a result of findings of occult metastasis [26].

Taking together all aspects of surgical strategies for HCC and CC forms the basis for the understanding of surgery for combined hepatocellular–cholangiocarcinoma (cHCC-CC). However, several non-surgical factors, such as tumor dimension, status of liver parenchyma and the extent of vascular invasion, are affecting surgical outcomes as well.

### 4.2. Liver Transplantation

The role of liver transplantation in combined hepatocellular–cholangiocarcinoma (cHCC-CC) remains controversial to date. The poor prognosis and the high recurrence rate of the tumor are generally regarded as contraindications for liver transplantation. Several retrospective studies have analyzed the results after liver transplantation for cHCC-CC with recurrence rates of approximately 40% [27,28]. Brandão et al. underwent a propensity-matched analysis of patients with HCC and cholangiocarcinoma and patients with cHCC-CC after liver transplantation. They found early tumor recurrence and lower recurrence-free survival (RFS) for patients with cholangiocarcinoma compared to patients with HCC or cHCC-CC. However, comparing the outcomes of liver transplant patients with HCC and cHCC-CC, there were no significant differences. These results indicate a subgroup of selected patients with cHCC-CC may benefit from liver transplantation [29]. A recent multi-center analysis included 208 patients with cHCC-CC after liver transplant and resection and compared the results with patients after liver transplantation and resection for HCC. They found that liver transplantation patients for cHCC-CC and HCC within Milan criteria showed no significant difference in overall survival despite higher cHCC-CC recurrence rates (23.1% vs. 11.5% after 5 years, *p* < 0.001) [30]. Regardless of tumor burden, outcomes after liver transplantation are superior to resection for a subgroup of patients with cHCC-CC. Within Milan criteria, liver transplantation for cHCC-CC and HCC resulted in similar overall survival. These results justify the consideration of liver transplantation due to the higher chance of cure in this traditionally excluded population.

### 4.3. Lymph-Node Dissection

To obtain information concerning the metastatic status and prognosis of the tumor, a perihepatic lymphadenectomy should be considered, although the therapeutic value is unproven. The incidence of positive lymph nodes is high, with some studies showing nodal disease in more than 40% of patients with intrahepatic cholangiocarcinoma [31]. However, to date, there is neither consensus on the indication nor on the extent of a perihepatic lymphadenectomy for cHCC-CC. As the hepatoduodenal ligament is the most common site of lymph-node metastases, a regional lymphadenectomy removing lymph nodes in the portocaval, pancreaticoduodenal region and along the common hepatic artery up to the celiac axis should be performed. More extensive lymph-node dissections including the para-aortic lymph nodes are not indicated in general. The number of metastatic lymph nodes strongly correlates with long-term survival, suggesting that systemic lymph-node dissection can help to identify patients with high risk for tumor recurrence and may improve survival in patients with limited nodal involvement [32]. To date, there are no benchmarks to guide the ideal number of lymph nodes during lymphadenectomy [33]. However, in case of distant lymph-node metastasis or grossly positive porta hepatis lymph nodes, the outcome is poor. Therefore, for patients with distant lymph-node involvement, systemic chemotherapy followed by restaging should be discussed as a meaningful concept [34].

### 4.4. Vascular Resections

Tumor infiltration of hepatic veins or the portal vein is one of the strongest prognostic factors in liver tumors. Vascular invasion is therefore a contraindication to liver transplantation. For extrahepatic cholangiocarcinoma of the hilum, data suggest that selected patients may benefit from portal vein resection [35]. What does that mean for combined hepatocellular–cholangiocarcinoma? Due to the intrahepatic localization of most combined hepatocellular–cholangiocarcinomas (cHCC-CC), extrahepatic portal vein infiltration is rare. On the other hand, gross invasion of hepatic veins is more frequent in patients with cHCC-CC. Survival data for HCC patients after hepatectomy with resection of hepatic veins for primary liver tumors showed a 3-year survival rate of 50%. Patients after resection of the inferior vena cava showed a 5-year survival rate of 33% [36]. However, these data focused on cholangiocarcinoma, since data concerning vascular resection for cHCC-CC is rare. Resection of the inferior vena cava therefore should be considered to achieve tumor-free resection margins and to offer the prospect of prolongation of survival in patients with otherwise poor prognosis. However, the mortality and complication rates are significantly increased after major vascular resections [37]. Therefore, it has to be emphasized that major vascular resections are technically complex and should only be undertaken in specialized centers.

In summary, the best surgical approach should be selected based on the current state of knowledge for cHCC-CC:

Explorative laparoscopy should be performed in advance of liver resection to exclude peritoneal carcinomatosis.

Anatomical and major hepatectomies should be preferred to remove residual satellite tumor cells and micro-tumors of the same anatomical area.

In order to realize a resection at all, even in advanced tumor stages with vascular invasion, resection of portal vein and hepatic vein can be considered.

Regional lymph-node dissection should be performed to gain oncological radicality.

### 4.5. Role of Perineural Invasion

Perineural invasion (PNI) describes tumor cell invasion through the perineurium. PNI, as a process with various histological features and underlying molecular mechanisms, is known as a significant risk factor for poor prognosis [38]. Recent data demonstrate that PNI is an independent risk factor even in R0-resected patients with intrahepatic cholangiocarcinoma [39]. However, perineural invasion seems to play a minor role in hepatocellular carcinoma. Recent data elucidating the prognostic relationship between prognosis and clinicopathological parameters of HCC, CC and combined hepatocellular–cholangiocarcinoma (cHCC-CC) showed 40% perineural invasion in patients with CC, whereas patients with HCC and cHCC-CC did not show any signs of perineural invasion [40]. Data comparing intrahepatic cholangiocarcinoma and perihilar cholangiocarcinoma showed histopathological parameters, such as portal vein invasion, lymphatic invasion and nodal metastasis, more common in intrahepatic cholangiocarcinoma. The incidence of perineural invasion was similar for both entities [41]. Compared to CC and perihilar cholangiocarcinoma, the perineural invasion status seems not to play an essential role in cHCC-CC and has no significance as a prognostic factor.

### 4.6. Surgical Outcome

Combined hepatocellular–cholangiocarcinoma (cHCC-CC) is associated with poor long-term survival [42]. Data regarding long-term survival varies amongst different studies. A review of the relevant literature shows that many studies compare the long-term oncological outcome of cHCC-CC with that of HCC and CC, most of them being comparative, retrospective analyses. An analysis by Gentile et al. included 13,613 patients with a diagnosis of cHCC-CC, CC or HCC treated with surgery [43]. They found a disease-free survival (DSF) of 15% after 5 years for patients with cHCC-CC compared to 31.6% for HCC and 20.3% for cholangiocarcinoma. The overall-survival (OS) was 32.7% for cHCC-CC, 47.5% and 30.3% for patients with HCC and CC, respectively. A subgroup analysis of the cohort compared the long-term oncological outcomes after liver transplantation for cHCC-CC and HCC and found a significantly worse DFS and OS after liver transplantation for cHCC-CC (5-year DFS was 40.9% for cHCC-CC versus 87.4% for HCC). However, the authors did not stratify within Milan criteria and therefore the results may show a bias to the disadvantage of cHCC-CC. For patients after liver resection, both DFS and OS of cHCC-CC were not statistically different from those of patients with cholangiocarcinoma (5-year DFS was 15% versus 20.3% and OS was 32.7% vs. 30.3%, *p* = 0.065). A study by Song et al. compared the prognosis and clinicopathological data of cHCC-CC and CC after resection. They found a median recurrence-free survival of 0.9 years for patients with cHCC-CC compared to 1.3 years for patients with CC (*p* = 0.028) [44]. Tumor number and vascular invasion have been identified as independent risk factor for recurrence-free survival in both groups. The size of the tumor seems to play an important role. For patients with CC, a cut-off size of 5cm was significantly associated with median survival. Interestingly, for patients with cHCC-CC, a cut-off size of 2 cm was associated with increased median disease-free survival [45]. A study from the Asan Medical Center reviewed the medical records of 29 patients with cHCC-CC. Disease-free survival at 6 months, 1 year and 3 years was 51.1%, 38.3% and 25.6%, respectively. They identified high levels of CA 19-9 and TNM—stages significantly associated with DSF [45]. Yang et al. identified 286 patients with cHCC-CC and 780 patients with CC from the Surveillance, Epidemiology, and End Result cancer database (SEER) and evaluated OS and cancer-specific survival (CSS). The 1-year, 3-year, and 5-year OS was 47.1%, 25.7% and 21.3% for patients with cHCC-CC. Overall survival probabilities in the CC group were 46.4%, 19.6% and 13.8%. The 1-year, 3-year and 5-year CSS probabilities were 52.1%, 30.5% and 26.9%, respectively, in the cHCC-CC group and 49.2%, 22.3% and 16.3%, respectively, in the CC group. The 1-year overall survival and CSS were comparable between the two groups, but the 5-year survival was significantly lower for patients with cholangiocarcinoma (*p* < 0.05) [46]. When analyzing prognostic factors for survival outcomes, they identified tumor size, M1 stage, American Joint Committee on Cancer (AJCC)—stage IIIc and IV, as well as surgery and chemotherapy, to be significantly associated with overall survival.

However, due to the rarity of this tumor prognosis, deterring clinicopathological characteristics remain poorly understood. Several studies have shown that most recurrences occur early after hepatic resection. Early recurrence rates of patients with cHCC-CC are about 57–75% and similar to patients with CC and higher than patients with HCC [47,48]. Focusing on patients with very early recurrence (recurrence within 6 months after initial resection, (VER)), recurrence rates of about 40% are reported by Ma et al. [49]. To identify cHCC-CC patients at risk for VER after surgical resection, Wu et al. validated a nomogram for prediction of VER. They included 131 patients and found microvascular invasion, macrovascular invasion and elevated carbohydrate antigen (CA 19-9) as independent risk factors of VER in the multivariate regression analysis. They constructed and evaluated a VER nomogram integrating microvascular invasion, macrovascular invasion and CA 19-9 > 25 U/mL to predict early tumor recurrence after liver resection [50]. This kind of tool may help to identify patients at risk of early tumor relapse and to adjust surgical indication and the extent of liver resection.

The main prognostic factors for cHCC-CC are large tumor size (>5 cm), the presence of satellite nodules, lymph-node involvement, multifocality of the tumor, portal vein invasion, vascular invasion, higher tumor stages and high levels of CA 19-9 [51]. Furthermore, free surgical margins < 2 cm are essential to increase prognosis [45]. This is of particular importance, especially for the surgical concept and the definition of the extent of liver resection. Recently Lin et al. proposed a new substaging system able to provide more precise prognosis for Barcelona clinic Liver Cancer Stage C hepatocellular carcinoma using tumor size > 10 cm, extrahepatic spread, macrovascular invasion, and Child-Pugh class [52]. This substaging system could also be useful in the cHCC-CC setting.

### 4.7. Role of Minimally Invasive and Robotic Surgery

Liver surgery for primary and secondary malignancies of the liver has long been conventional surgery exclusively. However, minimally invasive surgery, also known as minimally invasive liver surgery (MILS), has been established in the last 10–15 years [53,54]. The surgical significance of laparoscopic liver surgery is not clear yet in general. Many open questions remain, such as whether major hepatectomies including hemihepatectomy of the right or left lobe are feasible with comparative results to open surgery. Furthermore, the feasibility of minimally invasive lymphadenectomy at the liver hilum remains demanding since a regional lymphadenectomy for cHCC-CC is crucial. Data concerning surgical radicality are still rare. However, a recent analysis from Korea showed promising results after laparoscopic liver resection for cHCC-CC. The authors included 135 patients with cHCC-CC. A total of 43 patients underwent laparoscopic liver resection and 92 patients open liver resection. The 3-year overall survival was 88.4% in the laparoscopic group and 91.3% in the open surgery group with shorter hospital stay for patients after minimally invasive liver resection. The authors concluded that minimally invasive liver surgery for cHCC is technically feasible and safe [55]. However, to date laparoscopic major liver resections cannot be seen as a general standard and should only be performed in centers with excellent expertise in both liver surgery and minimally invasive surgery.

The introduction of robotic-assisted surgery during the last 10 years has revolutionized the field of minimally invasive surgery. Robotic surgery has been established to overcome the disadvantages of conventional laparoscopic surgery. The role of robotic-assisted liver surgery has not been evaluated yet in general. Nevertheless, robotic-assisted surgery is undoubtedly one of the most innovative advancements in terms of surgical techniques in recent years. Due to the significant advances in visualization and the minimally invasive character of laparoscopic procedures with optimized instrument mechanics, robotic surgery seems to possess the ability to make surgery more precise, especially highly complex procedures such as liver surgery [56,57].

Systemic data regarding robotic-assisted surgery for cHCC-CC are rare to date. We found one case from Brazil reporting a 73-year-old male undergoing robotic-assisted central hepatectomy for cHCC-CC. The postoperative period was reported as uneventful and the patient was discharged on the fifth postoperative day, indicating that robotic-assisted surgery for cHCC-CC is feasible and basically an alternative to conventional or laparoscopic liver surgery, even in older patients [58]. For liver surgery in general, a consensus is emerging that robotic liver surgery is medically meaningful. However, valid scientific data are still scarce, and financial aspects remain controversial [59]. It is to be expected that the role of robotic liver surgery will increase prospectively, similar to colorectal surgery or pancreatic surgery. The first results of studies comparing open to robotic liver resections seem to push away the laparoscopic role more and more. For the implementation of robotic liver surgery, the generally easier learning curve for robotic compared to laparoscopic surgery is an important factor [60,61].

## 5. Role of Chemotherapy

The role of adjuvant chemotherapy for cHCC-CC patients is still a grey zone. Sorafenib has been approved as a first-line treatment for advanced HCC and gemcitabine in combination with cisplatin for unresectable cholangiocarinoma. Due to the rare incidence of the tumor, no data concerning adjuvant therapy are available to date [62]. Therefore, data on long-term survival after curative intended resection are at least questionable since no standardized systematic therapies have been established.

In summary, survival rates of cHCC-CC are lower compared to HCC, suggesting that the clinical behavior of cHCC-CC is closer to CC than to HCC. It therefore appears that in cHCC-CC, the tumor proportion of cholangiocarcinoma determinates the outcome. Long-term oncological outcomes after liver resection and liver transplantation for cHCC-CC are shown in Table 1 and Table 2.

## 6. Conclusions

Combined hepatocellular–cholangiocarcinoma (cHCC-CC) is an aggressive primary liver tumor associated with poor long-term oncological outcomes. The tumor shares features of both HCC and CC and stands for a separate tumor entity. Its biological behavior, however, is much closer to CC than to HCC. Therefore, the proportion of cholangiocarcinoma seems to determinate the outcome. Surgical resection is the only curative treatment option. Oncologic liver surgery according to the current state-of-the art should be applied to improve the poor prognosis. Anatomical and major hepatectomies should be preferred to gain oncological radicality and to remove residual satellite tumor cells of the same anatomical area. Multidisciplinary approaches should be considered to realize resections even in advanced tumor stages with vascular invasion. Although there is no reliable data on the role of lymph-node dissection, a regional lymph-node dissection should be performed to get prognostic information and gain oncological radicality. Higher tumor stages, vascular invasion and high levels of CA 19-9 could be identified as significant prognostic factors. Since surgical treatment is the only curative option, liver transplantation should also be evaluated in highly selected cases. Data with regard to overall survival after liver transplantation for cHCC-CC are thoroughly promising. The roles of minimally invasive and robotic surgery in cHCC-CC are not clear yet. However, complex cases and major liver resections might be excellent candidates for robotic liver surgery with minimal concomitant damage and visual and instrumental augmentation.

## Figures and Tables

**Table 1 cancers-15-00774-t001:** Comparative, retrospective analysis of long-term oncological outcomes after liver resection for cHCC-CC.

Author	Year	Patients (*n*)	DFS (Years) 1 3 5	OS (Years) 1 3 5	Median OS (Months)
Yano Y et al. [3]	2003	26	- - -	- 34.6% 23.1%	21
Lee JH et al. [5]	2011	30	TTR (time to recurrence rate)	63.3% - -	18.3
Tang D et al. [63]	2006	14	- - -	84.6% 50.1% 50.1%	60
Yin X et al. [64]	2012	113	49.8% 13.2% 4.4%	73.9% 41.4% 36.4%	16.5
Lee SD et al. [65]	2014	42	46.5% 32.5% 32.5%	80.2% 61.3% 54.1%	61.9 mean
Yoon YI et al. [66]	2016	53	- - -	73.3% 35.6% 30.5%	8 after tumor recurrence
Li Z et al. [67]	2017	35	- - -	51% 36% 26%	24.9
Lin CW et al. [68]	2021	35	74.3% 28.6% 5.7% (calculated)	88.5% 62.2% 44.0%	50.1
Tang Y et al. [69]	2021	135	52.6% 37% 17% (calculated)	64% 42.9% 19.3% (calculated)	20.5

**Table 2 cancers-15-00774-t002:** Comparative, retrospective analysis of long-term oncological outcomes after liver transplantation for cHCC-CC.

Author	Year	Patients (*n*)	DFS (Years) 1 3 5	OS (Years) 1 3 5	Median OS (Months)
Sapisochin G et al. [28]	2014	15	Cumulative risk of recurrence	93% 78% 78%	-
Lunsford K et al. [70]	2018	12	66% 42% 42%	75% 54% 42%	-
Chang C et al. [71]	2017	11	80% 46.7% 46.7%	90% 61.7% 41.1%	-
Sapisochin G et al. [72]	2011	14	Cumulative risk of recurrence (HCC-CC and I-CC)	76% 65% 51%	-

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
