# Peer review of "Surgical Strategies for Combined Hepatocellular-Cholangiocarcinoma (cHCC-CC)"

_cancers, 2023, doi:10.3390/cancers15030774_

Round 1

Reviewer 1 Report

In this study, Ali Yahyazadeh et al address that Surgical strategies for combined hepatocellular-cholangiocarcinoma (cHCC-CC). This study reviewed surgical strategies for cHCC-CC.

The review study is interest and relevance and there is some space for improvement. In particular, as presented the study looks too descriptive in nature with some insights for the observations made.

Minor comment:

1.     The tumor behavior of cHCC-CC is some similar with HCC or CC. Surgical resection is the curative therapy for cHCC-CC. Surgical strategies for HCC and CC forms the basis for the understanding of surgery for cHCC- CC. However, several non-surgical factors such as tumor dimension, status of liver parenchyma and the extent of vascular invasion are affecting the surgical outcome also. Our previous study showed the new substaging system provides more precise prognosis to better tailor therapy for BCLC-C HCC patients using tumor size ≥10 cm, extrahepatic spread, macrovascular invasion, and Child-Pugh class. Surgical resection had better prognosis. The new substaging system may be useful for the prognosis and surgical strategies for cHCC-CC (Lin CW et al. Resubclassification and clinical management for Barcelona Clinic Liver Cancer Stage C hepatocellular carcinoma. Hepatol Int. 2021 Aug;15(4):946-956.). Please discuss it.

2.     Table 1. The 3-year DFS is 28.6%.

Author Response

Dear sir or madame,

thank you very much for considering our paper and also for your kind suggestions for improvement.

We hereby want to demonstrate our corrections:

The tumor behavior of cHCC-CC is some similar with HCC or CC. Surgical resection is the curative therapy for cHCC-CC. Surgical strategies for HCC and CC forms the basis for the understanding of surgery for cHCC- CC. However, several non-surgical factors such as tumor dimension, status of liver parenchyma and the extent of vascular invasion are affecting the surgical outcome also. Our previous study showed the new substaging system provides more precise prognosis to better tailor therapy for BCLC-C HCC patients using tumor size ≥10 cm, extrahepatic spread, macrovascular invasion, and Child-Pugh class. Surgical resection had better prognosis. The new substaging system may be useful for the prognosis and surgical strategies for cHCC-CC (Lin CW et al. Resubclassification and clinical management for Barcelona Clinic Liver Cancer Stage C hepatocellular carcinoma. Hepatol Int. 2021 Aug;15(4):946-956.). Please discuss it.

The new substaging system has been added to the manuscript.

  1. Table 1. The 3-year DFS is 28.6%.

 The table has been revised. 

Attached you will find the current version of the manuscript with the "track changes" function. Again we would like to thank your for your comments and we hope that we now fullfil your requirements. 

With best regards 

Marcus Bahra and Ali Yahyazadeh

Reviewer 2 Report

Dr Bahra and Yahyazadeh present a review that assesses therapeutic strategies combined HCC-CC

Overall, the paper is nicely interesting, however, I’d like to highlight some concerns regarding the manuscript:

-          Consider providing some TC/MRI imagines illustrating radiological features of combined HCC-CC

-          It results unclear authors’ point of view on liver biopsy for patients with HCC-CC. Could the authors comment further on that?

-          However, for surgical re-section of HCC the indication has been expand over the last to decades.”. The term to, should probably be changed to “two”.

-          A FLR ratio of 40% in healthy liver looks a bit arbitrary. Several publications have endorsed a FLR of 30% or even 20% in young patients with a healthy liver. Please, consider this issue.

-          Authors make some mention of PVE, however, it would probably be more appropriate to make references to two staged hepatectomies as liver deprivation, ALPPS, etc.. should be included as well in the group of staged hepatectomy.

-          The issue of anatomical vs non-anatomical resection for HCC and iCC is still a matter of debate. I’d recommend softening the tone about the advantages of anatomical resection (including references to 2 and 2-5 cm cut-offs)

-          The is a paragraph on adjuvant chemotherapy in the “surgical outcome sections”. I’d recommend to create a new and different section.

-          “systematic data” on page 8 should be corrected. “according to the current should” on page 8 should be corrected as well. Overall, some proofreading from a native English speaker would be recommended. 

Author Response

Dear sir or madame,

thank you very much for considering our paper and also for your kind suggestions for improvement.

We hereby want to demonstrate our corrections:

 Consider providing some TC/MRI imagines illustrating radiological features of combined HCC-CC.  Due to the fact, that a special chapter  concerning radiological features of combined HCC-CC  is planned in this special issue, we would like to entrust this topic for a contributing author specialized in radiology.

It results unclear authors’ point of view on liver biopsy for patients with HCC-CC. Could the authors comment further on that? The problems concerning multilocular biopsies ist adressed in the current version of the manuscript.

 “However, for surgical re-section of HCC the indication has been expand over the last to decades.”. The term to, should probably be changed to “two”. This part has been corrected.

A FLR ratio of 40% in healthy liver looks a bit arbitrary. Several publications have endorsed a FLR of 30% or even 20% in young patients with a healthy liver. Please, consider this issue. We have specified the FLR ratio in the manuscript.

Authors make some mention of PVE, however, it would probably be more appropriate to make references to two staged hepatectomies as liver deprivation, ALPPS, etc.. should be included as well in the group of staged hepatectomy. This topic has been added in the current version of the manuscript.

The issue of anatomical vs non-anatomical resection for HCC and iCC is still a matter of debate. I’d recommend softening the tone about the advantages of anatomical resection (including references to 2 and 2-5 cm cut-offs). This part has been modified in the current version of the manuscript.

The is a paragraph on adjuvant chemotherapy in the “surgical outcome sections”. I’d recommend to create a new and different section. A new chapter has been created as requested.

 “systematic data” on page 8 should be corrected. “according to the current should” on page 8 should be corrected as well. Overall, some proofreading from a native English speaker would be recommended.  As requested parts of the text were corrected an an additional proofreading has been performed by an english native speaker.

Attached you will find the current version of the manuscript with the "track changes" function. Again we would like to thank your for your comments and we hope that we now fullfil your requirements. 

With best regards 

Marcus Bahra and Ali Yahyazadeh
